# Core outcome sets for spinal and associated limb, trunk, abdomen or pelvic pain: A systematic review

Tim Noblet[1,2,3]*, Carol Li[4], Richard Newsham-West[5], David Walton[2], Alison Rushton[1,2]

**1** St George's University Hospitals NHS Foundation Trust, London, United Kingdom, **2** School of Physical Therapy, Western University, London, Ontario, Canada, **3** Faculty of Medicine, Health and Human Sciences, Lifespan Health and Wellbeing Research Centre, Macquarie University, Australia, **4** Faculty of Health Sciences, Western University, London, Ontario, Canada, **5** Australian and New Zealand College of Anaesthetists (ANZCA), Melbourne, Australia

* timnoblet@hotmail.com

## Abstract

### Background

Spinal pain is a significant global health issue, affecting millions and ranking as one of the leading causes of disability worldwide. Despite the wide scope of research conducted on spinal and associated pain, the lack of standardised core outcome measures poses challenges for comparing and synthesising research data. Core Outcome Sets (COSs) are intended to harmonise assessment and facilitate comparison across studies. This review aimed to identify, map, and examine published core outcome sets (COSs) designed for the assessment of spinal pain—including cervical, thoracic, lumbar—and spinal-related limb, trunk, abdomen, or pelvic pain. It also sought to synthesise consistent outcome domains across these COSs, categorising them by anatomical region and measurement type, including patient-reported, physical, biological, psychological, social, and environmental measures.

### Methods

This systematic review followed PRISMA guidelines and was registered with PROSPERO. A comprehensive literature search of 13 electronic databases and grey literature sources was conducted from 2000 to April 2025. Two independent reviewers assessed study eligibility and quality using predefined criteria. Data extraction was performed to identify core outcome domains, and a thematic analysis was conducted to categorise domains based on anatomical regions, patient-reported outcomes, performance measures, and biopsychosocial factors.

**Data availability statement:** All relevant data are explicitly tabulated within the paper and its Supporting Information files.

**Funding:** The author(s) received no specific funding for this work.

**Competing interests:** The authors have declared that no competing interests exist.

## Results

Thirteen studies met inclusion criteria, addressing core outcome sets for cervical (n = 4), thoracolumbar (n = 1), and lumbar (n = 8) spinal regions. Patient-reported outcome measures were the most frequently recommended outcome type. The most commonly endorsed domains were physical function n = 9 (100%), pain intensity n = 8 (88.9%), participation in work or daily activities n = 7 (77.8%), and disability n = 6 (66.7%). However, few studies incorporated psychological, social, environmental, or physiological domains, highlighting critical gaps in the multidimensional assessment of spinal pain.

## Conclusion

This systematic review identified key domains in current use and significant gaps in biopsychosocial and biological measurement. Findings will support researchers, clinicians, and policymakers in selecting appropriate outcomes for spinal pain research and practice. A Delphi study to develop an internationally agreed "Essential Universal Set" for spinal pain, inclusive of multidimensional biopsychosocial domains, is a sound next step.

---

## Introduction

Spinal pain is a complex and multifaceted health issue, with diverse aetiologies, clinical presentations, and treatment approaches [1]. On a global scale, approximately 790 million people experience spinal pain along with associated pain in the limbs, trunk, abdomen, or pelvis at any given time [2]. Spinal pain has persistently ranked as the leading cause of disability spanning from adolescence to older adulthood, with the number of disability-affected years rising worldwide and has consequently been identified as the leading need for rehabilitation intervention [2–5]. The costly consequences of spinal pain, both in terms of health and economics, are exacerbated by low-value care, inefficient and fragmented care pathways, and the absence of a person-centred approach [6–8]. These costs are projected to escalate further in the coming decades [2,6,8].

The research investigating spinal pain displays remarkable diversity in its approaches, methodologies, and areas of focus [6]. Researchers and practitioners from various disciplines, including medicine, physical therapies, psychology, neuroscience and public health, have contributed to the extensive body of knowledge surrounding spinal pain [1,6,8–10]. This research encompasses a broad range of topics, such as the aetiology, epidemiology, diagnostic methods, treatment modalities, rehabilitation techniques, and psychosocial factors influencing pain experience and management [8,9,11–13]. Moreover, studies explore different populations, including adults, adolescents, and older adults, as well as specific subgroups like athletes or those with specific spinal conditions [1,6]. Research methods vary, including clinical trials and observational studies, systematic reviews, qualitative interviews, and other

designs [1,14,15]. This diversity in research can be helpful for providing a comprehensive understanding of spinal pain, enabling the development of tailored interventions and improved outcomes for people affected by this complex condition [1,2,6].

The diversity of research also poses challenges in comparing and synthesising data, thereby limiting the ability to draw robust conclusions and make evidence-based decisions [6,16–18]. The lack of standardisation in outcomes particularly hampers meaningful comparison, data pooling, and the development of clinical guidelines. In an effort to harmonize outcomes across studies, several Core Outcome Sets (COSs) have been published intended for use in research on spinal pain. COSs consist of standardised measurements that capture the essential domains of interest in a specific health condition, such as spinal pain [16–19]. These COSs enable consistent evaluation of interventions and facilitate comparison across studies of, for example, treatment effectiveness. By promoting research quality, reliability, and transparency, COSs stand to contribute significantly to advancements in the understanding and management of spinal pain and associated pain conditions [9,16–20].

Several core outcome sets (COSs) have been developed for spinal pain and related musculoskeletal conditions over the past two decades. Notable examples include outcome sets for low back pain [21,22] whiplash-associated disorders [23] and spinal deformity surgery [24]. However, these efforts have generally focused on isolated anatomical regions or specific conditions, leading to fragmentation in measurement approaches. To date, no systematic review has comprehensively mapped and appraised COSs across the full spectrum of spinal pain and its common referred pain presentations,

Despite several relevant COSs being published, there has yet to be widespread consensus on the use of any one set. This poses a problem for both research and clinical practice in the field of spinal pain [9,16–20]. To tackle these issues, this study aims to systematically collate, examine, and evaluate the existing COSs employed in assessing spinal and associated pain,

### Aim

To identify what core outcome sets are currently used in the assessment of spinal pain and associated limb, trunk, abdomen, or pelvic pain, and to thematically analyze the domains to identify consistencies and gaps across them.

### Objectives

1. To identify, map and examine published core outcome sets intended for use in the assessment of: spinal (cervical, thoracic, lumbar) pain; spinal related lower limb pain; spinal related upper limb pain; and spinal related trunk, abdomen or pelvic pain.

2. To synthesise the consistent domains across core outcome sets for spinal pain and associated limb, trunk, abdomen, or pelvic pain. For easier interpretation, domains are conceptualized and categorized according to anatomical region of relevance, mode of implementation (patient-reported, physical/performance-based, or clinician/observer-rated) and broad construct (e.g., biological, physiological, psychological, social, environmental).

### Methods

This systematic review was conducted according to a pre-defined protocol with pre-specified eligibility criteria, in order to identify and analyse empirical research relating to the study's objective [25]. This methodology was selected as a systematic review aims to critically appraise and synthesise results, providing an explicit overview and detailed map of the current evidence [26–28]. To maximise the confidence of findings and minimise bias, the protocol was informed by the Cochrane handbook [25,29–31] to facilitate the completion of low risk of bias systematic review by using recommendations for appropriate methodological decision-making informed by empirical evidence [25]. For clarity and transparency, this review

is reported in accordance with the PRISMA statement (Preferred Reporting Items for Systematic Reviews and Meta-analyses) developed by a panel of international experts to ensure the quality of reporting [31,32].

## Protocol and registration

The systematic review is registered with PROSPERO (CRD 42023473529), a register of anticipated health and social care systematic reviews held by the University of York Centre for Reviews and Dissemination. Registration ensures transparency, reducing bias caused by potential selective reporting and was essential to avoid duplication [29,33].

## Eligibility criteria

An adapted PICOS framework ((P) patient population or the disease being addressed, (I) interventions or exposure, (C) comparator/control group, (O) outcome measure, (S) study design)) was used to inform eligibility criteria [25,34]. PICOS enables specific elements of clinical evidence to be identified in a systematic review [25,34,35]. The eligibility criteria are detailed in Fig 1.

## Information sources

The literature search employed sensitive topic-based strategies designed for each of the sources identified in Fig 2. The specific databases searched were selected following discussion with subject matter experts, methodologists, and advice from specialist librarians at St George's University of London (UK) and Western Univerity (Ontario, Canada). The information resources accessed were chosen to enable a holistic literature search [25,29,36]. Grey literature was explored using ProQuest Dissertations and Theses, along with trial registers such as clinicaltrials.gov and the International Clinical Trials Registry Platform of the World Health Organization. In addition, experts in the area were consulted to detect any further studies [25,36–38].

## Search strategy

Pre-defined search terms and combinations, with database specific standardised vocabulary were employed to ensure all relevant studies were retrieved [25,36–38]. This was essential to ensure that no studies were missed due to different words being used to describe the same concepts [25,36–38]. The search of databases encompassed the period over the 25-year window from 01.01.2000 to 01.04.2025 to ensure all contemporary COSs were captured. Fig 3 demonstrates the MEDLINE (Ovid) search strategy that was adapted to meet search terms of other included databases. Where a protocol or pilot study was identified, the definitive study was sought, or the authors were contacted to determine whether further published or unpublished research has been undertaken. The reference lists of the included studies were searched to ensure no studies were missed [36,38].

## Data management

A web-based software platform (COVIDENCE, Veritas Health Innovation, Melbourne Australia) was used for managing various stages of systematic and other literature reviews. This platform facilitated the importation of all citations, assisted in eliminating duplicates arising from the search strategy, and supported the determination of eligibility throughout the screening and review phases [39].

## Selection process

Two investigators searched the information sources (TN + CL) and independently assessed studies for inclusion by grading each eligibility criterion. The use of two investigators reduced the possibility of missing relevant literature or excluding relevant research where challenging judgements were necessary for selection or rejection of a study

**Inclusion criteria:**

**Population:**
Core outcome sets (COSs) developed for use in adults (aged 18–70) experiencing spinal or spinal-related pain of musculoskeletal origin, including (18-70):
- Low back pain and low back-related leg or pelvic pain
- Thoracic spine pain and thoracic-related trunk, abdominal, or pelvic pain
- Cervical spine pain and cervical-related upper limb pain

**Concept:**
Development or consensus-based refinement of COSs intended to guide the assessment or evaluation of community-based adult populations with non-cancer spinal pain and with a focus on rehabilitation outcomes. This includes outcome sets focusing on domains or constructs, whether or not specific measurement instruments are recommended.

**Context:**
COSs applicable to any setting (primary, secondary, or tertiary care; clinical or research), delivered in public or private healthcare systems, and developed in any geographical location or professional discipline.

**Types of Sources:**
Eligible sources included primary research studies, systematic reviews, meta-analyses, and clinical guidelines that describe the development or structure of COSs related to spinal or spinal-related pain.

**Outcome:**
Inclusion of at least one defined outcome domain or construct (e.g. pain, function, quality of life). Inclusion was not dependent on the presence of specific outcome instruments (21).

**Exclusion criteria:**
Publications that only describe regional or cultural adaptations of existing COSs to avoid double-counting sets/domains. We also excluded core outcome measure sets specifically intended for central nervous system disorders (e.g., stroke/post-stroke pain, spinal cord injury, concussion/traumatic brain injury, cerebral palsy, etc.), disorders driven by viral infections (e.g. post-herpetic neuralgia, HIV/AIDS-induced pain, etc.), cancer-related pain (e.g. metastases) or treatment of cancer (e.g. chemotherapy-induced pain), pain disorders specifically related to pregnancy complications (e.g. pre-eclampsia), endometriosis, or menstrual pain, pain conditions driven by endocrine or hormonal disorders (e.g. hyper/hypocortisolism, adrenal insufficiencies, etc.), pain disorders related to systemic immune or inflammatory processes (e.g. rheumatoid arthritis, ankylosing spondylitis, etc.), and core outcome measure sets focused on surgical outcomes (e.g., complications/failure/revision rates, length of stay, etc.).

**Fig 1. Eligibility criteria.**

Databases searched:

CINAHL, EMBASE, MEDLINE, AMED, NHS Economic Evaluation database, NICE, Medicines Complete, Cochrane Collaboration databases, ProQuest,PUBMED

Grey Literature & selected internet sites:
Turning Research into Practice, website (York), Google Scholar, The World Health Organisation, COMET.org, OMERACT, clinicaltrials.gov, ICHOM.org

National Research Register

Expert Opinion

Hand searches- key journals

**Fig 2. Information sources.**

[33]. In the event of a selection disagreement a third reviewer (AR/DW, methodological experts) was consulted [33,37]. Both reviewers independently evaluated studies by title and abstract for potential eligibility. Following discussion between reviewers, if a study could not be explicitly excluded based on its title and abstract, its full text was reviewed [29,31]. All potentially relevant studies proceeded forward to full text review after which the two reviewers made independent judgements as to whether an individual study met the eligibility criteria. Any conflict was resolved with a 3rd reviewer. The numbers of studies included and excluded at the different stages was recorded using a PRISMA flow diagram [25,33,36].

### Data collection process

Data extraction and charting was performed by the primary reviewer (TN) and checked and agreed by the secondary reviewer (CL). Data collection used pre-determined data collection sheets specific to the review objectives which were piloted, refined and agreed by the researchers prior to use [33,36]. Any differences were resolved at a consensus meeting of all researchers [37], and the third reviewers (AR/DW) checked for consistency and clarity.

### Data items and outcomes

The following data items and outcomes were extracted from all included studies: authors, date of publication, study aim(s)/objective(s) and design, anatomical region of interest, definition of the condition the COS was used for, recommended outcome measures included within the COS, including verbatim domains or constructs (or specific tools) endorsed and recommendations for the frequency of use for patient assessment [25,27,36].

### Quality (including internal validity) assessment of individual studies

As of this review there were no widely-accepted risk of bias (RoB) tools published for critical appraisal of Delphi or consensus research, which was anticipated to be the most commonly used method for development of the COSs. Accordingly, for purposes of this review we adapted the Conducting and Reporting of Delphi Studies (CREDES [40]) best practice framework to evaluate the quality (including internal validity) of the included studies regarding conduct and reporting [40].

| | |
|---|---|
| 1. | low back pain.mp. or exp Low Back Pain/ |
| 2. | LBP.mp. |
| 3. | exp Sciatica/ or low back related leg pain.mp. |
| 4. | LBLP.mp. |
| 5. | exp Radiculopathy/ or lumbar radicular.mp. |
| 6. | radicular pain.mp. |
| 7. | low back related pelvic pain.mp. |
| 8. | exp Abdominal Pain/ or abdominal pain.mp. |
| 9. | exp Cervical Vertebrae/ or cervical spine pain.mp. or exp Neck Pain/ |
| 10. | exp Lumbar Vertebrae/ |
| 11. | neck pain.mp. or exp Neck Pain/ |
| 12. | exp Pelvic Pain/ |
| 13. | Cx spine.mp. |
| 14. | Tx spine.mp. |
| 15. | exp Thoracic Vertebrae/ or thoracic spine.mp. |
| 16. | mid back pain.mp. |
| 17. | thoracic spine trunk pain.mp. |
| 18. | cervical radiculopathy.mp. or exp Radiculopathy/ |
| 19. | 1 or 2 or 3 or 4 or 5 or 6 or 7 or 8 or 9 or 10 or 11 or 12 or 13 or 14 or 15 or 16 or 17 or 18 |
| 20. | outcome measure*.mp. or exp "Outcome and Process Assessment, Health Care"/ |
| 21. | self reported outcome measure.mp. |
| 22. | exp Patient Reported Outcome Measures/ |
| 23. | core data set*.mp. |
| 24. | 20 or 21 or 22 or 23 |
| 25. | 19 and 24 |
| 26. | exp Delphi Technique/ or exp Consensus/ or delphi.mp. |
| 27. | guidelines.mp. or exp Guideline/ or exp Practice Guideline/ |
| 28. | framework*.mp. |
| 29. | 26 or 27 or 28 |
| 30. | 25 and 29 |

**Fig 3. Medline (Ovid) search strategy.**

While other frameworks exist for evaluating COS development and reporting—namely COS-STAD (Core Outcome Set–Standards for Development) [41] and COS-STAR (Core Outcome Set–Standards for Reporting) [42]—these were not used directly in this review. COS-STAD provides methodological guidance for COS design, and COS-STAR outlines standards for transparent reporting. Despite their distinct focuses, both frameworks share key domains such as clear rationale, stakeholder engagement, consensus methodology, and scope definition. The CREDES checklist was chosen over COS-STAD and COS-STAR because it integrates domains relevant to both frameworks and is specifically tailored to Delphi and consensus methodology [40–42]. This made it the most suitable tool for appraising both the design and reporting quality of the included studies, capturing aspects of internal validity, transparency, and methodological consistency across the consensus-based COS literature.

CREDES was developed by methodological experts to enable quality and consistency in the conduct, reporting and appraisal of Delphi studies, using a 16-item checklist [40] specifically tailored to Delphi and consensus methodology [40–42]. A modified version of the 'Buchbinder appraisal scale' (BAS) was used to evaluate each included study against the 16-items within the CREDES [43–45]. The BAS awards one point for fully meeting a criterion, half a point for partially meeting a criterion, or zero points for not meeting a criterion or being unable to score due to a deficit in the information reported [43–45]. Therefore, a quality score between 0–16 was reported of each of the included studies [40]. The BAS was originally developed for the critical appraisal of treatment based classification systems across 7 criteria with an overall Intraclass Correlation Coefficient (ICC) of 0.82 for inter-rater reliability [43,44]. The scale has now been adopted successfully across a range of studies where quality assessment tools are yet to be developed and validated [43,44].

## Synthesis of results

A three-step process was used to synthesise the data.

**Stage 1: Identification and mapping of COSs**

An explanation of each included study's characteristics and COS outcomes/recommendations were tabulated.

**Stage 2: Synthesis of outcome measures across regions of the spine**

For each anatomical region, data across the studies were synthesised to identify the assessment meaures recommended in the following categories: patient reported measures (PROMs), physical/ performance-based measures, biological measures, clinician-assessed psychological measures, social measures, environmental/ ecological measures [26–28]. A narrative synthesis was used to synthesise the measures within spinal pain and spinal related limb, chest, abdominal, and pelvic pain 'regions' [27,29,37,46].

**Stage 3: identification of recurrent outcome domains across COSs**

The grouping of data across studies and spinal region into core domains, was completed by using the tabulated data from stages 1 and 2. The constructs endorsed within each core outcome measure set were grouped through thematic content analysis [47]. This process started by reviewing and re-reviewing the verbatim constructs included within outcome measures within each COS, and where specific PROMs were endorsed, returning to the literature as needed to define the construct(s) being measured by each. Domains were interpreted verbatim per the COS authors, while one-off or customized COS-specific questions or questionnaires were interrogated for the domain(s) with which they most aligned based either on the original development publications of those tools or, where the question or tool lacked adequate theoretical grounding in its initial development, interpreted against contemporary domains of health and wellness with the International Classification of Functioning, Disability, and Health (ICF) as the primary framework. Each domain, question, or PROM were therefore aligned with one or more dominant domains for thematic analysis.

Following extraction and interpretation of the most accurate construct from all measures included in the COSs, the thematic grouping moved from specific to broad, wherein groups of like constructs were clustered together to form 'Sub-Domains' that continue to be clustered into conceptually-distinct 'Domains' with consensus reached by all authors. A domain was considered to be in "common use" if it was reported in more than two thirds (66.6%) of the included core

outcome sets. This threshold was selected based on its use in previous quality studies assessing the frequency of domains within patient-reported outcome measures (PROMs) [48].

### Certainty of evidence

Assessment of the overall certainty or confidence in the evidence was not conducted in this systematic review. Established frameworks such as GRADE (Grading of Recommendations Assessment, Development and Evaluation) [49,50] and GRADE-CERQual (Confidence in the Evidence from Reviews of Qualitative Research) [51] were not appropriate for the included studies, as they are primarily designed for assessing certainty in intervention studies and qualitative evidence, respectively. The studies included in this review were primarily Delphi and consensus-based, which lack a suitable, validated framework for assessing certainty of evidence [49–51]. As such, no formal assessment of certainty was applied.

## Results

### Study selection

The search returned n = 8,046 potentially relevant studies. Following screening for duplicates, n = 5,790 citations remained. One additional study was identified from the Internet searches, and one from hand-searching the citation lists of the remaining articles, both of which were subsequently screened. Reviewing by title and abstract excluded n = 5,751 studies that clearly did not meet the eligibility criteria. No relevant unpublished studies were found, and no further studies were identified from reviews of the national research register or via experts in the field. The full texts of the remaining n = 39 studies were examined in detail and evaluated against the eligibility criteria.

Two studies—White et al (2004) [52] and Rebbeck et al (2007) [53]—were 'whiplash associated disorder' (WAD)- and 'mechanical neck pain'-specific adaptations of a COS developed by Deyo et al (1998) [21] for low back pain. While the Deyo publication (1998) pre-dated publication cut-off of 2000, it was retained in the final selection of sources as the two subsequent adaptations were otherwise uninterpretable without the context of the original. All three were individually rated for quality and included separately in stages 1 and 2 of the data synthesis. In stage 3, they were considered a single COS (with region-specific adaptations) for data extraction and synthesis, as their constructs were consistent. Similarly, Verburg et al (2019) [24] and Verburg et al (2021) [54] were two studies from one programme of research. Both met eligibility criteria and were retained. Again, they were rated and synthesised separately in stages 1 and 2, but treated as a single COS in stage 3 as both retained the same outcome domains although the later study introduced outcome-based quality indicators for patient-reported outcome measures (PROMs). For the same reasons Chen et al (2019) [55] and Sterling et al (2023) [23] were also considered a single study in stage 3, as Sterling et al (2023) [23] provided specific measurement instruments for the COS constructs initially defined by Chen et al (2019) [55].

Thirteen studies were included in the systematic review. 100% inter-rater agreement was achieved following open discussion at each stage. Third reviewer mediation was not required. Fig 4 presents the number of studies at each stage of the selection process.

Stage 1: Identification and mapping of core outcome measure sets

### Study characteristics

Studies were published between 1998–2023. Study designs included Delphi technique (n = 5) [22,24,55–57], other consensus methods (n = 6) [21,23,54,58–60], and adaptations and evaluation of existing COS (produced through consensus methods) to produce region-specific COS (n = 2) [52,53].

Four studies described COSs for the cervical spine region [23,52,53,55]: three focused on whiplash-associated disorders [23,53,55] and one on mechanical neck pain [52]. Eight studies described COSs for the lumbar spine region exclusively [21,22,24,54,56,58–60]. Of these, four addressed 'nonspecific' or 'mechanical'

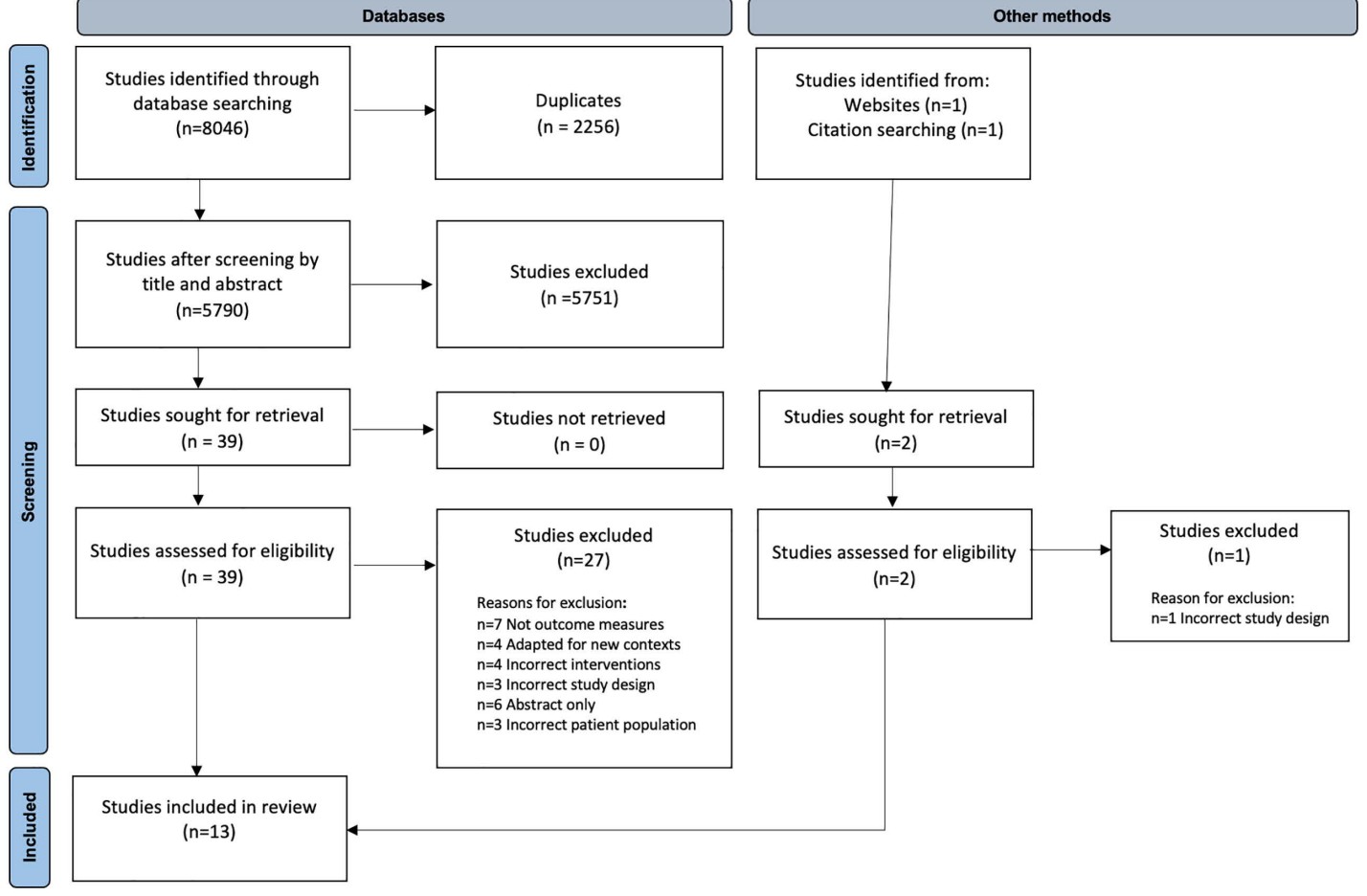

**Fig 4. PRISMA flow diagram.**

low back pain [21,24,54,58], two were intended for all low back pain excluding cases with sinister pathology, inflammatory disease, infection, or fracture [22,56], one focused exclusively on chronic low back pain [59], and one was inclusive of both acute and chronic low back pain [60]. One study investigated the thoracolumbar region (including both thoracic and lumbar regions) [57], specifically targeting adolescent and young adult patients with spinal deformity.

Six studies developed COSs specifically for research purposes [21–23,55,59,60], seven for clinical practice [24,52–54,56,57], and one for both research and clinical settings [58]. COSs developed for clinical practice were generally shorter, including fewer domains or measures, whereas those designed for research settings tended to be more comprehensive, incorporating a broader array of domains or measures.

In terms of scope, two studies aimed to define core outcome domains only [55,58], one focused on defining appropriate patient-reported outcome measures (PROMs) [24], and ten defined both outcome domains and the PROMs to assess them [21–23,52–54,56,57,59,60].

Only three studies explicitly defined the frequency with which the COS should be used [24,56,60]. One study recommended specific application strategies tailored to individual studies [59], while the remaining nine did not specify frequency of use [21–23,52–55,57,58].

## Quality assessment

Quality assessment data can be found in Supplementary file 1. Scores ranged from 0.5 to 15 out of a maximum of 16 points. The most commonly met quality domains across studies included clear justification for using the Delphi or consensus technique, well-defined consensus procedures, and adequate interpretation of results. High-scoring studies consistently addressed these criteria. In contrast, the least commonly met domains were prevention of bias, external validation, and detailed reporting of expert panel and participant recruitment.

## Results of individual studies

The data and quality assessment outcome from individual studies (COSs) are presented in Table 1.

## Results of syntheses

Stage 2: Synthesis of outcome measures.

Table 2 presents a synthesis of outcome measures across spinal regions, structured according to the outcome domains reported within each included study. *Each outcome is categorised under the domain as defined by the original authors of the respective core outcome set.*

## Cervical spine COSs

Three studies [23,52,53] with a quality rating of 0.5 to 14/16 reported outcome meaures in COSs for the cervical spine. All three COSs endorsed standardized PROMs. None included physical/performance-based, biological, non-patient-reported psychological, social, or environmental/ecological measures.

**Pain** was the most commonly included domain, appearing in all three COSs [23,52,53]. The Visual Analogue Scale (VAS) was included for use in all three. Two [52,53] (quality 0.5/16) included a custom symptom scale, while one [23] (quality 14/16) included the Numeric Pain Rating Scale (NPRS).

**Disability** was included in two COSs [52,53] (both quality 0.5/16) using tailored questions about daily activity limitation and time lost from work or school due to symptoms.

**Percieved Recovery** (global percieved change) was included in one COS [23] (quality 14/16) using a 7- or 11-point Global Rating of Change (GROC) scale.

**Physical function** was included in one COS [23] (quality 14/16) using both the Neck Disability Index (NDI) and the Whiplash Disability Questionnaire (WDQ).

**Psychological functioning** was included in one COS [23] (quality 14/16), which recommended the use of validated instruments including the Pictorial Fear of Activity Scale-Cervical (PFActS.-C.), the Pain Self-Efficacy Questionnaire, and the Pain Catastrophising Scale.

**Satisfaction with care** was included in one COS [52] (quality 0.5/16) using a single-item measure regarding the participant's overall impression of their medical care.

**Social function** was included in all three COSs [23,52,53]. Two C.O.S. [52,53] (both quality 0.5/16) included a 'PROM question' that asked how pain interfered with work and home responsibilities. One C.O.S. [23] (quality 14/16), included two 'PROM questions', asking about employment status (e.g., modified duties, disability pension) and the percentage of usual work hours performed.

**Wellbeing** assessment was included in two COSs [52,53] (quality 0.5/16) using a single global item asking participants how they would feel if their current symptoms persisted for the rest of their lives.

## Thoracic spine COSs

One study [57] with a quality rating of 12.5/16 reported a COS for the thoracic spine comprised entirely of standardized PROMs. The Scoliosis Research Society–22 revised questionnaire (SRS-22r) was the primary PROM included, capturing

**Table 1. Data from individual studies.**

| Study | Aim & Design | Region | Definition of Condition | Core Outcome Measure Sets | | | Frequency of Use | Quality /16 |
|---|---|---|---|---|---|---|---|---|
| | | | | Domains | Sub-domains | Measures | | |
| 1. CHEN (2019) [55] | Aim: To develop a set of core outcome domains recommended for use in all **clinical trials** for WAD. Design: 3-step process (1)Identification of outcome domains via a literature survey. (2)3-round Delphi survey of relevant stakeholders to reach consensus on the identified core domains. (3)Multidisciplinary consensus meeting to finalise the domains. Participants: Delphi- n= 93 Researchers, health care providers, patients, and compensation insurance personnel). Consensus meeting- n= unknown (International Steering Committee, clinical researchers, health care providers, insurance and patient representatives. | Cervical | WAD were defined as the clinical manifestations (e.g., neck pain, headache, dizziness) that arise as a result of an acceleration-deceleration mechanism of energy transfer to the neck. It may result from rear-end or side-impact motor vehicle collisions, but can also occur during diving or following falls. WAD grades I to III were the focus. | 6 core domains: •Physical Functioning •Perceived Recovery •Work & Social Functioning •Psychological Functioning •Quality of Life •Pain | | Not investigated | Not defined | 15 |
| 2. CHIAROTTO (2018) [22] | Aim: To develop recommendations for core outcome measurement instruments for use in **clinical trials** involving patients with non-specific low back pain (nsLBP). Design: 4-step process (1)Identification of potential core outcome measurement instruments by steering Committee from those frequently used in clinical trials and recommended by initiatives aimed at standardising measurements for LBP. (2)Appraisal of measurement properties of outcome measurement instruments. (3)Online modified Delphi survey. (4)Steering committee consensus. Participants: Steering committee n= 19 anesthesiology, epidemiology, internal medicine, orthopaedics, physical therapy, neurosurgery, primary care, psychology, rehabilitation, rheumatology and patient representatives. Dephi: n= 207 (response rate, round 1, 44%, round2, 41%). Authors of at least 2 publications comprising psychometric or clinimetric studies, randomised clinical trials, or systematic reviews of clinical trials in patients with nsLBP. | Lumbar | LBP not attributable to a recognisable known specific pathology, e.g., infection, tumour, fracture, and axial spondyloarthritis. | 5 core domains: •Function •Pain Symptoms •Generic Health Status •Work Disability •Satisfaction with Care | | Function: ODI 2.1a (if able to fund fees), otherwise RMDQ-24. Pain Symptoms: NRS. HRQoL: SF12 (if able to fund fees), otherwise PROMIS-GH-10. In clinical trials with economic evaluation EQ-5D-5L to be included. | Not defined | 15 |

*(Continued)*

Table 1. (Continued)

| Study | Aim & Design | Region | Definition of Condition | Core Outcome Measure Sets | | Measures | Frequency of Use | Quality /16 |
|---|---|---|---|---|---|---|---|---|
| | | | | Domains | Sub-domains | | | |
| 3. CIEZA (2004) [58] | Aim: To report on the results of the consensus process integrating evidence from preliminary studies to develop the first version of the ICF Core Sets for LBP: the Comprehensive ICF Core Set and the Brief ICF Core Set. Use in **research and clinical practice.** Design: •A formal consensus process integrated evidence from a Delphi exercise, systematic review, and empirical data collection using the ICF checklist. •Trained international experts identified relevant ICF categories based on evidence and expert judgment. Participants: n=18 (14 with various physicians sub-specialisations, 3 occupational therapists & 1 physical therapist) from 15 different countries. The decision-making process for LBP involved 3 working groups, with 6 experts, respectively. | Lumbar | Low back pain. | 4 core domains: •Body functions •Body structures •Activities & participation •Environmental factors | Body functions: *Sensation of pain, emotional functions, muscle power functions, mobility of joint functions, exercise tolerance functions, sleep functions, muscle endurance functions, muscle tone functions, stability of joint functions, energy and drive functions)* Body structures: *spinal cord & related structures, structure of trunk, additional musculoskeletal structures related to movement)* Activities & participation: *maintaining a body position, lifting & carrying objects, changing basic body position, walking, remunerative employment, work & employment, doing housework,dressing, handling stress & other psychological demands, family relationships, toileting, acquiring, keeping & terminating a job)* Environmental factors: *health services, systems & policies, social security services, systems and policies, health professionals, individual attitudes of health professionals, individual attitudes of immediate family members, products & technology for employment, products or substances for personal consumption, immediate family, design, construction and building products & technology of buildings for private use, legal services, systems and policies* | Not investigated | Not defined | 2.5 |
| 4. CLEMENT (2015) [56] | Aim: To define a universal, internationally applicable set of outcomes for standard **clinical practice** in low back pain care. Design: Modified Delphi process involving 6 teleconferences. First round proposals were based on a review of the academic literature, review of existing practices in spine registries and direct input from working group members and other experts in the field. Participants: Delphi: n=22 (surgical, rehabilitation, & medical experts in the field of low back pain, plus 1 expert patient). | Lumbar | Low back pain including: lumbar spinal stenosis, lumbar spondylolisthesis, degenerative disc disorders including disc hemiation, degenerative scoliosis, other degenerative lumbar disorders, and acute and chronic lumbar back pain and back-related leg pain without a clear etiology (e.g., mechanical or nonspecific pain). | 5 core domains: •Pain •Disability •Quality of life •Work Status •Analgesic use | | Pain: NRS Disability: ODI Quality of life: EQ5D-3L; EQ-VAS Work status: Question •*"what is your current work status?"* Analgesic use: Question •*Do you take non-narcotic pain relieving medications or tablets for your back problem? Yes regularly, yes sometimes, no.* | Baseline, index event(s), 6 months, 1 year, 2 years.s | 7.5 |

*(Continued)*

**Table 1.** (Continued)

| Study | Aim & Design | Region | Definition of Condition | Core Outcome Measure Sets | | | Fre-quency of Use | Quality /16 |
|---|---|---|---|---|---|---|---|---|
| | | | | Domains | Sub-domains | Measures | | |
| **5. DEYO (1998) [21]** | Aim: To promote more standardization of outcome measurement in **clinical trials and other types of outcomes research**, including meta-analyses, cost-effectiveness analyses, and multicenter studies. Design: An international group of back pain researchers considered recommendations for standardised measures in clinical outcomes research in patients with back pain. The panel considered several factors in recommending a standard battery of outcome measures. These included reliability, validity, responsiveness, and practicality of the measures. In addition, compatibility with widely used and promoted batteries such as the American Academy of Orthopaedic Surgeons Lumbar Cluster were considered to minimise the need for changes when these instruments are used. Participants: The panel: n= not reported. Defined as a multinational group of investigators. | Lumbar | Low back pain. | 6 core domains: •Pain symptoms •Function •Well being •Disability •Disability-social role •Satisfaction with care | | Questions for use in both clinical practice and research: Pain symptoms: *"During the past week, how bothersome have the following symptoms been?" a) low back pain b) leg pain (sciatica)* Or Conventional visual analog pain scales Function: *"During the past week, how much did pain interfere with your normal work (including both work outside the home and housework?"* Well being: *"If you had to spend the rest of your life with the symptoms you have right now, how would you feel about it?"* Disability: *"During the past 4 weeks, about how many days did you cut down on the things you usually do for more than half of the day because of back pain or leg pain (sciatica)?"* Disability-social role: *"During the past 4 weeks, how many days did low back pain or leg pain (sciatica) keep you from going to work or school?"* Statisfaction with care: *"Over the course of treatment for your low back pain or leg pain (sciatica), how would you rate your overall medical care?"* Expanded set of core instruments for clinical researchers: Function: RMDS or ODI Well being: SF-12 or EuroQol Disability-social role: add questions about days of work absenteeism, days in which the patient had to cut down on normal & days spent in bed for at least half a day. | Not defined | 2 |

*(Continued)*

Table 1. (Continued)

| Study | Aim & Design | Region | Definition of Condition | Core Outcome Measure Sets | | | Fre-quency of Use | Quality /16 |
|---|---|---|---|---|---|---|---|---|
| | | | | Domains | Sub-domains | Measures | | |
| 6. DEYO (2015) [59] | Aim: To develop a set of standards for **clinical research** on cLBP. Objective of importance: •Report a minimum dataset for research studies. •Rank the importance of measures of pain-related behavioral, emotional, and psychosocial domains influencing the expression of cLBP. Design: e-mail surveys of the research task force (RTF) members. (1) Members ranked the importance of baseline descriptors for patients with cLBP, including medical history, comorbidities, physical examination, and laboratory/imaging tests. (2) Members ranked the importance of self-report measures for pain-related behavioral, emotional, and psychosocial domains affecting cLBP. Participants: RTF members: n=16 (chronic low back pain experts). | Lumbar | Chronic low back pain. | 4 core domains: •Physical function •Depression •Sleep disturbance •Catastroph-ising | | Minimum dataset Medical History: •Demographics •Involvement in workers' compensation or legal claims •Work status •Education •Comorbidity (*smoking status, obesity, substance abuseusing the* two-item conjoint scale & widespread pain symptoms) •Previous treatment history (history of surgical interventions & use of opioid analgesics) Physical Examination: •Straight leg raise (SLR) for those with low back related leg pain (LBLP) •Hip medial rotation •Lower limb muscle strength •Lower limb reflexes (for studies including radiculopathy) Diagnostic Testing: •MRI of the lumbar spine required forstudies of surgical interventions Self Report Measures: •Short form- PROMIS (For physical function, the ODI or RMDS, and for depression, the PHQ-9 or Beck Depression Inventory, were considered acceptable alternatives based on investigator preference). Adverse Events: No specific recommendations for reporting/measuring. Expanded measures: Studies focused on behavioural or mood correlations should incorporate additional measure- no specific instrument identified. | Specific to the individual study. | 5 |

*(Continued)*

Table 1. (Continued)

| Study | Aim & Design | Region | Definition of Condition | Core Outcome Measure Sets | | | Measures | Fre-quency of Use | Quality /16 |
|---|---|---|---|---|---|---|---|---|---|
| | | | | **Domains** | **Sub-domains** | | | | |
| 7. DE KLEU-VER (2017) [57] | Aim: to develop a **patient-relevant clinical** core outcome set for adolescent and young patients with spinal deformity, that will facilitate benchmarking within and between the 5 countries of the NSDS and other registries worldwide. Design: Modified Delphi study was performed, which consisted of a literature review (preparatory stage) and 4 consensus rounds. Prticipants: Delphi: n=7 (Spinal surgeons >5yrs spinal deformity specialty, representatives of the national spine surgery registries from each NSDS country (2 from Sweden, 2 from Denmark, and 1 each from Finland, Norway, and the Netherlands). | Thoracic/ Lumbar | Adolescent and young adults with spinal deformity requiring surgery fro associated symptoms, e.g.,pain. | 13 core domains: Body function and structures •Physical function •Pain intensity •Self-image •Change in deformity •Pulmonary fatigue Environmental outcomes •Re-operation •Complications Participation •Recreation and leisure Activities •Health related quality of life •Satisfaction with cosmetic result •Physical functioning •Pain interference •Satisfaction with overall outcome | | | Self-image, physical function-ing, pain interference, physical function, pain intensity, recre-ation & leisure, satisfaction with the cosmetic result, satisfaction with the overall outcome of surgery: SRS-22r, HRQoL: EQ-5D Pulmonary fatigue: no vali-dated measurement instrument was found that measures patient-reported for the target population. Re-operation, complications, change in deformity: clinician reported. | Not defined | 12.5 |

(Continued)

| Study | Aim & Design | Region | Definition of Condition | Core Outcome Measure Sets | | | Frequency of Use | Quality /16 |
|---|---|---|---|---|---|---|---|---|
| | | | | Domains | Sub-domains | Measures | | |
| 8. PINCUS (2008) [60] | Aim: Improve the quality of **prospective investigations [research]** into the transition from early stages of low back pain (LBP) to persistent problems by recommending an agreed minimal list of measures for inclusion in baseline data collection. Primarily aimed at researchers who want to investigate prognosis in LBP, allowing data from cohorts to be pooled to compare between different health care systems. Design: Consensus study including 4 stages: 1. Generation of factors (domains) for consideration from current literature & team workshops. 2. Consensus for inclusion or exclusion of factors (domains): team consensus activity and expert comments. 3. Selection of measures: Literature review. 4. Consensus for inclusion of measures: Team approval activity. Participants: Teams: x11 international teams- experts in low back pain. Steering Group: n=8 (experts in low back pain). Systematic review: n=2 (members of the steering group). | Lumbar | Acute & chronic low back pain. | 8 core domains: •Disability and well-being •Healthcare utilisation, including referral and treatment •Medication •Pain •Patient satisfaction with care/condition •Tests and examinations including bloods, FMRI, radiographs •Work •Quality of live | | Wellbeing: SF36 Disability: RMDS Healthcare utilisation: number of consultations/referals with/ for: Consultation with behavioral therapist: counselor/psychologist/ pain management/ manual therapists: physiotherapist, osteopaths, chiropractors/ Medical: neurologist, rheumatologist, GP, MDTs, Occupational health. Medication: use of analgesia Pain: NPS and question: "During the past 3 months, about how many days did you cut down on things you usually do for more than half the day because of back pain or leg pain?" Satisfaction with care: question "Over the course of treatment for your back pain or leg pain, how satisfied were you with your overall medical care?" Satisfaction with condition: question "If you had to spend the rest of your life with the symptoms you have right now, how would you feel about it?" Tests & examinations: Yes/No questions, bloods, FMRI, radiographs. Return to work: question "If you were off work due to back pain or leg pain, are you now 1) back to the same job; 2) back, but job modified to accommodate pain; 3) in a new job, more suited to accommodate pain; 4) not working because of pain; 5) not working or other reasons (use list from baseline)." Sick leave (days off due to back pain): question "During the past 3 months, how many days did low back pain or leg pain keep you from going to work or school?" Catastrophising: PCS Fear avoidance: FABQ Depression/destress: CES-D | Medications & Healthcare utilisation: every 2-weeks All other measures: every 12-weeks | 5 |

*(Continued)*

| Study | Aim & Design | Region | Definition of Condition | Core Outcome Measure Sets | | Measures | Fre-quency of Use | Quality /16 |
|---|---|---|---|---|---|---|---|---|
| | | | | Domains | Sub-domains | | | |
| 9. REBBECK (2007) [53] | Aim: To comprehensively evaluate the psychometric properties of a 5-item version of the Core Outcome Measure in people with whiplash. For use in **clinical practice.** Design: Data were sourced from 3 separate whiplash cohorts (total 481) encompassing acute, early chronic, and late-chronic whiplash among primary care and insurance populations. Subjects completed a 5-item version of the Core Outcome Measure for whiplash (Core Whiplash Outcome Measure [CWOM]), the Functional Rating Index, Neck Disability Index, SF-36, and perceived recovery questionnaires at baseline and short and long-term follow-up periods. Psychometric evaluation of the CWOM included assessing questionnaire responses, internal consistency, construct validity, and internal and external responsiveness. Participants: CWOM- modified questions from Deyo et al 1998/ White et al 2004. Questions were modified and agreed by the authors n = 4. | Cervical | Whiplash Associated Disorder (WAD). | 5 core domains: •Pain symptoms •Function •Well being •Disability •Disability-social role | | Questions for use in both clinical practice and research: Pain symptoms: *"During the past week, how bothersome have the following whiplash symptoms been?"* Or Conventional visual analog pain scales Function: *"During the past week, how much did pain interfere with your normal work (including both work outside the home and housework?"* Well being: *"If you had to spend the rest of your life with the symptoms you have right now, how would you feel about it?"* Disability: *"During the past 4 weeks, about how many days did you cut down on the things you usually do for more than half of the day because of whiplash symptoms?"* Disability-social role: *"During the past 4 weeks, how many days did whiplash symptoms keep you from going to work or school?"* | Not defined | 0.5 |

*(Continued)*

| Study | Aim & Design | Region | Definition of Condition | Core Outcome Measure Sets | | Measures | Frequency of Use | Quality /16 |
|---|---|---|---|---|---|---|---|---|
| | | | | Domains | Sub-domains | | | |
| 10. STERLING (2023) [23] | Aim: To make recommendations on core outcome measurement instruments for each of the 6 core domains identified in stage 1 (Chen et al 2019) for use in **clinical trials and cohort studies** of WAD (Physical Functioning, Perceived Recovery, Work and Social Functioning, Psychological Functioning, Quality of Life, and Pain). Design: 2-step process •Systematic reviews •Steering committee consensus activity Participants: Steering committee: n=18 (researchers in the area of WAD, 3 continents, n=5 undertook clinical work, n=1 patient representative. Expertise: physiotherapy, psychology, anaesthesiology, pain medicine, chiropractic, emergency medicine, epidemiology, health economics, internal medicine, orthopaedics, legal practice, rehabilitation, social work). | Cervical | WAD were defined as the clinical manifestations (e.g., neck pain, headache, dizziness) that arise as a result of an acceleration-deceleration mechanism of energy transfer to the neck. It may result from rear-end or side-impact motor vehicle collisions, but can also occur during diving or following falls. WAD grades I to III were the focus | 6 core domains: •Physical Functioning •Perceived Recovery •Work & Social Functioning •Psychological Functioning •Quality of Life •Pain | | Physical Functioning: NDI, WDQ Perceived Recovery: 7-point or 11-point GROC scale Work & Social Functioning: There were no studies investigating the validity of any single-item questions of work-related functioning in patients with WAD. Recommend 2 broad items. 1.determine current work (paid/unpaid, e.g., caring role, student) status. For example, regular duties, modified duties, work retraining because of injury, disability pension, not working because of injury, and not working forother reasons. 2.determine the percent of usual work (paid/unpaid as above) hours being performed. Psychological Functioning: No specific recommendation from the systematic review. Authors recommended 1 of 3 PROMs-PFActS-C, Pain Self-Efficacy Questionnaire, Pain Catastrophising Scale. If measuring PTSD, the Harvard Trauma Questionnaire or the Posttraumatic Stress Diagnostic Scale could be considered. Quality of Life: Owing to a lack of studies, the systematic review concluded that, despite being endorsed as an important outcome, no specific PROM recommendation could be made. Pain: NPRS and VAS | Not defined | 14 |

*(Continued)*

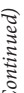

| Study | Aim & Design | Region | Definition of Condition | Core Outcome Measure Sets | | Measures | Frequency of Use | Quality /16 |
|-------|--------------|--------|------------------------|---------------------------|--|----------|------------------|-------------|
| | | | | Domains | Sub-domains | | | |
| **11. VERBURG (2019) [24]** | Aim: To develop a clinical standard set of outcome measures for low back pain in Dutch **primary care physiotherapy practices,** that are accepted for relevance and feasibility by stakeholders and useful for (a) interaction between patient and the professional, e.g., shared decision-making in goal-setting, monitoring and feedback based on outcomes, (b) internal quality improvement, and (c) external transparency in patients with non-specific low back pain (NSLBP) in primary care physical therapy. Design: Consensus-driven modified RAND-UCLA Delphi method in seven steps: 1.Literature search 2.First online survey 3.Patient interviews 4.Experts meeting 5.Consensus meeting 6.Second online survey 7.Final approval of an advisory board Participants: Delphi: n=32 out of 43 pannellists (patients, representatives of patient and physiotherapy associations, researchers, policy makers, health insurers, age 42–73yrs, 75.8% male). Advisory group: n=5 (n=2 private health insurers, n=2 physiotherapists, n=1 expert patient). | Lumbar | Non-specific low back pain. | Not formally defined. | | For patients classified as low-risk by STarT Back: •Patient-Specific Functional Scale (PSFS) •Numeric Pain Rating Scale (NPRS) •Global Perceived Effect—Dutch Version (GPE-DV) For patients classified as Medium/high-risk by STarT Back: •Quebec Back Pain Disability Scale (QBPDS) •Oswestry Disability Index (ODI)(pilot) •Patient-Specific Functional Scale (PSFS) •Numeric Pain Rating Scale (NPRS) •Global Perceived Effect—Dutch Version (GPE-DV) | Low-risk: Intake & end of treatment. Medium/ high-risk: Intake, every 6 weeks & end of treatment. | 13 |
| **12. VERBURG (2021) [54]** | Aim: to define and select a core set of outcome-based quality indicators for patients with non-specific low back pain in Dutch **primary care physical therapy,** accepted by stakeholders on usability and perceived added value as a quality improvement tool. Design: 2 phase mixed methods approach in a sequential explanatory design: 1.Quality indicator identification and evaluation (estimated comparability and discriminatory characteristics using prospectively collected patient outcomes in a convenience cohort). 2.Stakeholder group consensus exercise (4 focus groups). Participants: Quality indicator identification: n=5 (authors). Stakeholder group: n=23 (n=19 physical therapists, n=4 researchers). | Lumbar | Non-specific low back pain. | 4 core domains: •Pain intensity •Physical activity •Physical function •Perceived effect of treatment | | Pain intensity: NPRS Physical activity: Patient-Specific Functional Scale (PSFS) Physical function: Quebec Back Pain Disability Scale (QBPDS) Perceived effect of treatment: Global Perceived Effect—Dutch Version (GPE-DV) | Not defined | 2.5 |

*(Continued)*

Table 1. (Continued)

| Study | Aim & Design | Region | Definition of Condition | Core Outcome Measure Sets | | | Measures | Fre-quency of Use | Quality /16 |
|---|---|---|---|---|---|---|---|---|---|
| | | | | Domains | Sub-domains | | | | |
| **13. WHITE (2004) [52]** | Aim: To validate a new and brief outcome measure for use with patients with mechanical neck pain. For use in **clinical practice.**<br><br>Design:<br>The back pain measure (Deyo et al 1998) was adapted to enable its use with patients with neck pain (The short Core Neck Pain Questionnaire (CNPQ)). Repeatability was assessed by using a 1-week test/retest on 104 patients who were enrolled in a neck pain trial. Validity was assessed by comparing the new questionnaire against other already well validated measures (i.e., the Neck Disability Index and a Visual Analogue Scale for pain) with n = 133 patients.<br>Participants: CNPQ- modified questions from Deyo et al 1998. Questions were modified and agreed by the authors n = 3. | Cervical | Mechanical neck pain. | 6 core domains:<br>•Pain symptoms<br>•Function<br>•Well being<br>•Disability<br>•Disability-social role<br>•Satisfaction with care | | | Questions for use in both clinical practice and research:<br>Pain symptoms:<br>*"During the past week, how bothersome have the following symptoms been?"* a) neck pain b) shoulder/arm pain<br>Or<br>Conventional visual analogue pain scales<br>Function:<br>*"During the past week, how much did pain interfere with your normal work (including both work outside the home and housework?"*<br>Well being:<br>*"If you had to spend the rest of your life with the symptoms you have right now, how would you feel about it?"*<br>Disability:<br>*"During the past 4 weeks, about how many days did you cut down on the things you usually do for more than half of the day because of neck pain?"*<br>Disability-social role:<br>*"During the past 4 weeks, how many days did neck pain keep you from going to work or school?"*<br>Satisfaction with care:<br>*"Over the course of treatment for your neck pain, how would you rate your overall medical care?"* | Not defined | 0.5 |

**Table 2. Synthesis of outcome measures across regions of the spine.**

**Cervical Spine**

| Outcome type | Outcome Domain | Outcome Measures | Number of COS | Studies (Study No as per Table 1) | Quality (Range/16) |
|---|---|---|---|---|---|
| Patient Reported Outcome measures (PROMs) | Disability | Question: "During the past 4 weeks, about how many days did you cut down on the things you usually do for more than half of the day because of whiplash symptoms?" Question: "During the past 4 weeks, how many days did whiplash symptoms keep you from going to work or school?" | 2 | 9,13 | 0.5 |
| | Pain | VAS | 3 | 9, 10, 13 | 0.5-14 |
| | | NPRS | 1 | 10 | 14 |
| | | Question: "During the past week, how bothersome have the following whiplash symptoms been?" | 2 | 9,13 | 0.5 |
| | Perceived Recovery | 7-point or 11-point GROC scale | 1 | 10 | 14 |
| | Physical Function | NDI | 1 | 10 | 14 |
| | | WDQ | 1 | 10 | 14 |
| | Psychological Functioning | PFActS-C | 1 | 10 | 14 |
| | | Pain Self-Efficacy Questionnaire | 1 | 10 | 14 |
| | | Pain Catastrophising Scale | 1 | 10 | 14 |
| | Satisfaction | Question: "Over the course of treatment for your neck pain, how would you rate your overall medical care?" | 1 | 13 | 0.5 |
| | Social Function | Question: "During the past week, how much did pain interfere with your normal work (including both work outside the home and housework?" | 2 | 9,13 | 0.5 |
| | | Question to determine current work (paid/unpaid, e.g., caring role, student) status. For example, regular duties, modified duties, work retraining because of injury, disability pension, not working because of injury, and not working forother reasons. | 1 | 10 | 14 |
| | | Question to determine the percent of usual work (paid/unpaid as above) hours being performed. | 1 | 10 | 14 |
| | Well being: | Question: "If you had to spend the rest of your life with the symptoms you have right now, how would you feel about it?" | 2 | 9,13 | 0.5 |
| Physical and Performance-based measures | | | 0 | – | – |
| Biological measures | | | 0 | – | – |
| Psychological measures (non-PROMs) | | | 0 | – | – |
| Social measures (non-PROMs) | | | 0 | – | – |
| Environmental/ Eccological measures. | | | 0 | – | – |

*(Continued)*

**Table 2.** (Continued)

| Cervical Spine | | | | | |
|---|---|---|---|---|---|
| **Outcome type** | **Outcome Domain** | **Outcome Measures** | **Number of COS** | **Studies (Study No as per Table 1)** | **Quality (Range/16)** |
| **Thoracic Spine** | | | | | |
| Patient Reported Outcome measures (PROMs) | Pain | SRS-22r | 1 | 7 | 12.5 |
| | Physical function | SRS-22r | 1 | 7 | 12.5 |
| | Quality of Life | EQ-5D | 1 | 7 | 12.5 |
| | Recreation & leisure, | SRS-22r | 1 | 7 | 12.5 |
| | Self-image | SRS-22r | 1 | 7 | 12.5 |
| (Post-operative only) | Overall outcome of surgery | SRS-22r | 1 | 7 | 12.5 |
| | Satisfaction with the cosmetic result | SRS-22r | 1 | 7 | 12.5 |
| Physical and Performance-based measures | | | 0 | – | – |
| Biological measures (Post-operative only) | Re-operation, complications, change in deformity | Clinician reported | 1 | 7 | 12.5 |
| Psychological measures (non-PROMs) | | | 0 | – | – |
| Social measures (non-PROMs) | | | 0 | – | – |
| Environmental/Ecological measures. | | | 0 | – | – |
| **Lumbar Spine** | | | | | |
| Patient Reported Outcome measures (PROMs) | Analgesic use | Question: *Do you take non-narcotic pain relieving medications or tablets for your back problem? Yes regularly, yes sometimes, no.* | 2 | 4, 8 | 5-7.5 |
| | Catastrophising | Pain Catastrophising Scale (PCS) | 1 | 8 | 5 |
| | Depression | Beck Depression Inventory (BDI-II) | 1 | 6 | 5 |
| | | Center for Epidemiological Studies – Depression scale (CES-D) | 1 | 6 | 5 |
| | | Patient Health Questionnaire-9 (PHQ-9) | 1 | 8 | 5 |

*(Continued)*

Table 2. (Continued)

**Cervical Spine**

| Outcome type | Outcome Domain | Outcome Measures | Number of COS | Studies (Study No as per Table 1) | Quality (Range/16) |
|---|---|---|---|---|---|
| | Disability | ODI | 2 | 4, 11 | 7.5-13 |
| | | Quebec Back Pain Disability Scale (QBPDS) | 2 | 11, 12 | 2.5-13 |
| | | RMDS | 1 | 8 | 5 |
| | | Question: "During the past 4 weeks, about how many days did you cut down on the things you usually do for more than half of the day because of back pain or leg pain (sciatica)?" Question: "During the past 4 weeks, how many days did low back pain or leg pain (sciatica) keep you from going to work or school?" Question: "Number of days off work" Question: "Number of days in which the patient had to cut down on normal activities" Question: "Number of days spent in bed for at least half a day" | 1 | 5 | 2 |
| | Healthcare Utilisation | Number of consultations/referals with/for: Consultation with behavioral therapist/ counselor/psychologist/ pain management/ manual therapists/physiotherapist/osteopaths/chiropractors/ Medical | 1 | 8 | 5 |
| | Fear Avoidance | FABQ | 1 | 8 | 5 |
| | Pain | NPS | 1 | 8 | 5 |
| | | NRS | 4 | 2, 4, 11, 12 | 2.5-15 |
| | | SRS-22r | 1 | 7 | 12.5 |
| | | VAS | 1 | 5 | 2 |
| | | Question:"During the past week, how bothersome have the following symptoms been?" a) low back pain b) leg pain (sciatica) | 1 | 5 | 2 |
| | | Question: "During the past 3 months, about how many days did you cut down on things you usually do for more than half the day because of back pain or leg pain?" | 1 | 8 | 5 |
| | Physical Function | ODI 2.1a | 3 | 2, 5, 6 | 2-15 |
| | | Patient-Specific Functional Scale (PSFS) | 2 | 11, 12 | 2.5-13 |
| | | RMDQ-24 | 3 | 2, 5, 6 | 2-15 |
| | | SRS-22r | 1 | 7 | 12.5 |
| | | Question: "During the past week, how much did pain interfere with your normal work (including both work outside the home and housework?" | 1 | 5 | 2 |
| | Quality of Life | EQ-5D | 3 | 2, 4, 7 | 7.5-15 |
| | | EQ-VAS | 1 | 4 | 7.5 |
| | | PROMIS-GH-10 | 2 | 2, 6 | 5-15 |
| | | SF12 | 1 | 2 | 15 |
| | Recreation & Leisure (pre and post-operative) | SRS-22r | 1 | 7 | 12.5 |

*(Continued)*

**Table 2.** (Continued)

**Cervical Spine**

| Outcome type | Outcome Domain | Outcome Measures | Number of COS | Studies (Study No as per Table 1) | Quality (Range/16) |
|---|---|---|---|---|---|
| | Satisfaction | Global Perceived Effect | 2 | 11, 12 | 2.5-13 |
| | | Question: "Over the course of treatment for your low back pain or leg pain (sciatica), how would you rate your overall medical care?" | 2 | 5, 8 | 2-5 |
| | | Question: "If you had to spend the rest of your life with the symptoms you have right now, how would you feel about it?" | 1 | 8 | 5 |
| | Self-image (pre and post-operative) | SRS-22r | 1 | 7 | 12.5 |
| | Wellbeing | EuroQol | 1 | 5 | 2 |
| | | SF-12 | 1 | 5 | 2 |
| | | SF36 | 1 | 8 | 5 |
| | | Question: "If you had to spend the rest of your life with the symptoms you have right now, how would you feel about it?" | 1 | 5 | 2 |
| | Work Status | Question: "what is your current work status?" | 1 | 4 | 7.5 |
| | | Question: "If you were off work due to back pain or leg pain, are you now 1) back to the same job; 2) back, but job modified to accommodate pain; 3) in a new job, more suited to accommodate pain; 4) not working because of pain; 5) not working or other reasons (use list from baseline)." | 1 | 8 | 5 |
| | | Question: "During the past 3 months, how many days did low back pain or leg pain keep you from going to work or school?" | 1 | 8 | 5 |
| (Post-operative only) | Overall Outcome of Surgery | SRS-22r | 1 | 7 | 12.5 |
| | Satisfaction with the Cosmetic Result | SRS-22r | 1 | 7 | 12.5 |
| Physical and Performance-based measures | Physical Examination | Hip medial rotation | 1 | 6 | 5 |
| | | Lower limb muscle strength | 1 | 6 | 5 |
| | | Lower limb reflexes (for studies including radiculopathy) | 1 | 6 | 5 |
| | | Straight leg raise (SLR) for those with low back related leg pain (LBLP) | 1 | 6 | 5 |
| Biological measures | Adverse Events | Clinician reported | 1 | 6 | 5 |
| | Diagnostic Testing | Bloods | 1 | 8 | 5 |
| | | MRI | 2 | 6, 8 | 5 |
| | | Radiographs | 1 | 8 | 5 |
| Psychological measures (non-PROMs) | | | 0 | – | – |
| Social measures (non-PROMs) | | | 0 | – | – |
| Environmental/ Ecological measures | | | 0 | – | – |

multiple outcome domains relevant to thoracic spine conditions. These included pain, physical function, recreation and leisure, self-image, and overall outcome of surgery. In addition, satisfaction with the cosmetic result was assessed post-operatively using the same instrument. The COS also included the EuroQoL-5D (EQ-5D) to assess overall quality of life. An extended version of this COS specific to surgical outcomes was also presented that included clinician-assessed outcomes such as re-operation rates, post-operative complications, and changes in spinal deformity. No other outcome types—such as physical or performance-based measures, non-patient-reported psychological or social measures, or environmental/ecological measures—were included,

**Lumbar spine COSs**

Nine studies (exclusively lumbar n=8, thoracolumbar n=1) [21,22,24,54,56–60] with a quality rating range of 2 to 15/16 reported COSs for the lumbar spine. All nine COSs included PROMs to assess multiple outcome domains. One COS [61] (quality 5/16) included physical/performance-based measures and two COSs [60,61] (both with quality rating 5/16) included biological measures (blood tests, MRIs and radiographs). None included non-patient-reported psychological or social outcomes, or environmental and/or ecological measures.

**Pain** was the most frequently reported outcome domain, included in seven COSs (quality 2.5-15/16). The Numeric Pain Rating Scale (NPRS) was included in four COSs [22,24,54,56] with quality scores ranging from 2.5 to 15/16. One C.O.S. [21] (quality 2/16) included the Visual Analogue Scale (VAS), one COS [60](quality 5/16) included the Numeric Pain Scale (NPS), and two COSs [21,60] (quality 2-5/16) included custom symptom-specific questions. One thorocolumbar scoliosis-specific COS [57] (quality 12.5/16) included the SRS-22r which includes a pain subscale as part of a multidimensional PROM.

**Disability** was included in five COSs [21,24,54,56,60] (quality 2 to 13/16). These included: The Oswestry Disability Index (ODI) in two [24,56] (quality 7.5-13/16), the Quebec Back Pain Disability Scale (QBPDS) in two [24,54] (quality 2.5-13/16), and the Roland-Morris Disability Questionnaire (RMDQ) in one COS [60] (quality 5/16). One C.O.S. [21] (quality 2/16) included single-item disability questions, asking about days spent in bed, missed work, or restricted normal activity due to back or leg pain.

**Physical function** assessment was included in six COSs [21,22,24,54,57,61] (quality 2-15/16), using both validated PROMs and COS-specific structured questions. The O.D.I. 2.1a was included in three COSs [21,22,61] (quality 2-15/16). The Patient-Specific Functional Scale (PSFS) was included in two COS [24,54] (quality 2.5-13/16), while the RMDQ was included in three [21,22,61] (quality 2-15/16). One COS [57] (quality 12.5/16) included the SRS-22r to assess physical functioning as part of a multidimensional PROM. One COS [21] (quality 2/16) also included a single question on pain interference with work and home responsibilities.

**Quality of life** was included in four COS [22,56,57,61] (quality 5-15/16). The E.Q.-5D. was included in three COSs [22,56,57] (quality7.5-15/16), while the EuroQol visual analogue scale (E.Q.-VAS) was included for use by one [56] (quality 7.5/16). The S.F.-12 was recommended by one COS [22] (quality 15/16). Two C.O.S.s [22,61] (quality 5-15/16) included the P.R.O.M.I.S.-GH-10.

**Wellbeing** was included in two COSs [21,60] (quality 2-5/16). One C.O.S. [21] (quality 2/16) included the EuroQol, the S.F.-12, and included a question asking how patients would feel if their current symptoms persisted for life. One C.O.S. [60] (quality 5/16) included the S.F.-36.

**Patient-reported Psychological Function** were included in two COSs. Both the Pain Catastrophising Scale and Fear-Avoidance Beliefs Questionnaire (FABQ) were included in one COS [60] (quality 5/16). Two C.O.S.s [60,61] (quality-both studies 5/16) included measures to evaluate depression: one [61] included the Beck Depsression Inventory (BDI-II) and Center for Epidemiological Studies – Depression scale (CES-D), while the other [60] included the Patient Health Questionnaire-9 (PHQ-9).

**Satisfaction** with care was included in four COSs [21,24,54,60] (quality 2.5-13/16). Two C.O.S.s [21,60] (quality 2-5/16) included a single-item satisfaction question. Two others [24,54] (quality 2.5-13/16) included the Global Perceived Effect scale.

**Work status** was recommended for inclusion in two COSs [56,60] (quality 5-7.5/16). One [56] (quality 7.5/16) included a general work status question, while one [60] (quality 5/16) included more detailed questions about return-to-work outcomes and days missed due to back or leg pain.

**Pre and post-operative** outcomes were included in one COS [57] (quality 12.5/16) recommending the SRS-22r to assess the overall outcome of surgery, satisfaction with cosmetic results, recreation and leisure, and self-image.

**Physical and performance-based measures** were included in one COS [61] (quality 5/16), being clinician-assessed hip medial rotation, lower limb muscle strength, deep tendon reflexes, and straight leg raise (SLR) for patients with radicular symptoms.

**Biological measures** were included in two COSs [61]. Both included metrics extracted from diagnostic testing including MRI. One [60] (quality 5/16) also included use of blood tests and radiographs where indicated.

**Reporting of adverse events** was included in one [61] (quality 5/16).

No included studies reported non-patient-reported psychological, social, or environmental/ecological measures.

Stage 3: identification of recurrent outcome domains.

A total of n=82 domains, questions, or PROMs were extracted from across all included COSs detailed in Table 1. Thematic content analysis categorised the 82 recommendations into 22 conceptual domains (Table 3). Of those, four domains were endorsed by >6 (66.66%) of the 9 COSs meeting the threshold for 'common use' status: patient rated physical function (100%), pain symptoms/Pain intensity (88.9%), participation - work or school (77.8%), and participation - activities of daily living(66.7%).

## Reporting biases

For all eligible studies included in this review, complete data were available from the published reports. No additional data were required, and no contact with study authors was necessary to obtain missing information. As such, the risk of reporting bias related to unavailable or selectively reported data was considered minimal.

## Discussion

This systematic review comprehensively identified and synthesised core outcome sets for spinal pain and associated pain in the limbs, trunk, abdomen, or pelvis. While previous literature has explored specific outcome measures for specific spinal regions or clinical contexts, this review uniquely brings together COSs across multiple anatomical regions and domains of assessment, offering a pan-spinal overview.

This review identified 13 eligible manuscripts describing COSs (and derivatives thereof) intended for use in the cervical, thoracic, or lumbar spinal regions. Across these, PROMs were the most frequently endorsed outcome type. Core domains, most notably pain intensity, physical function, and participation in work and daily activities, were each recommended in at least two-thirds of the identified COSs suggesting that these are 'essential' domains as endorsed by experts and, in some cases, representatives of the patient population.

These findings are closely aligned with those reported by Sabet et al. (2025), whose systematic review of 40 COSs for musculoskeletal conditions similarly highlighted pain and function as the most commonly endorsed domains [62]. This alignment reflects longstanding clinical and research priorities and is consistent with recommendations from established initiatives such as OMERACT and the COMET Initiative, which advocate for the standardisation of core domains in health outcomes research [18,19,63]. The concordance observed reinforces the central importance of these domains in musculoskeletal and spinal research and underscores the necessity of their inclusion in any future COS developed for spinal pain. The consistent prioritisation of domains such as pain intensity and physical function across COSs likely reflects the

**Table 3. Presence of each of the 22 thematically-classified Domains endorsed by each of the COSs.**

| Domain | DEYO et al, 1998 Lumbar (Clinical Researchers) | DEYO et al, 1998 Lumbar (Routine clinical use) | WHITE et al, 2004 Cervical | REBBECK et al, 2007 Cervical | CHIAROTTO 2018 Lumbar | CIEZA et al, 2004 Lumbar | CLEMENT et al, 2015 Lumbar | DEYO et al, 2015 Lumbar | De Keuver et al, 2017 Thoracic/Lumbar | PINCUS et al, 2008 Lumbar | STERLING 2023/ CHEN 2021 Cervical | Verburg et al 2019/2021 Lumbar | TOTAL [/ 9] |
|---|---|---|---|---|---|---|---|---|---|---|---|---|---|
| Patient rated physical function | X | | | | X | X | X | X | X | X | X | X | 9 |
| Pain symptoms/Pain intensity | X | | | | X | | X | X | X | X | X | X | 8 |
| Participation (work or school) | X | X | X | X | X | X | X | X | | X | X | | 7 |
| Participation (activities of daily living) | X | X | X | X | X | X | X | X | | X | | | 6 |
| Emotional/Psychological functioning | | | | | X | | X | X | | X | X | | 5 |
| Quality of life | X | | | | X | | X | | X | | X | | 5 |
| Generic health status | X | | | | X | | X | | | X | | | 4 |
| Pain affect | X | X | | X | | X | X | | | | | X | 4 |
| Satisfaction with condition | X | X | | X | | X | | X | | | X | | 4 |
| Body structure or function/past medical history | | | | | | X | X | | | X | | | 3 |
| Health resource utilisation | | | | | | | X | X | | X | | | 3 |
| Satisfaction with social activities | | | | | X | | | | X | | X | | 3 |
| Global perceived effects/perceived recovery | | | | | | | | | | | X | X | 2 |
| Number of deaths | | | | | X | | X | | | | | | 2 |
| Satisfaction with care | X | | X | | | X | | | | | | | 2 |
| Sleep disturbance | | | | | | | | X | | | X | | 2 |
| Adverse Outcome of Treatment | | | | | | | | X | X | | | | 2 |
| Environmental | | | | | | X | | | | | | | 1 |
| Fatigue | | | | | X | | | | | | | | 1 |
| Nature of the condition | | | | | | | | X | | | | | 1 |
| Pain Frequency | X | | | | | | | | | | | | 1 |
| Patient data | | | | | | | | X | | | | | 1 |

Key:

Domains above threshold for 'Common use' status.

Domains below the threshold for 'Common use' status.

***NB: Deyo, White, and Rebbeck; Sterling and Chen; and Verburg (2019/2021) COSs were each treated as single sets for domain extraction to avoid duplication.

long-standing biomedical orientation of spinal pain research and the preference for outcomes that are easily measured, clinically interpretable, and responsive to intervention. Conversely, the limited inclusion of psychological, social, and biological domains may result from practical constraints, disciplinary silos in COS development, and concerns regarding feasibility in clinical settings. These patterns suggest that while core domains remain stable, broader biopsychosocial factors, essential to precision and person-centred care, are underrepresented, limiting the comprehensiveness of existing COSs.

Variation in the validated outcome tools used across the included COSs was evident, with several studies employing the same instrument to measure different domains. This was particularly apparent between the domains of physical function and disability, where tools were often applied beyond the scope for which they were originally developed or psychometrically validated. Such inconsistencies raise concerns regarding construct validity and comparability across studies.

The prominence of domains such as pain and physical function across both musculoskeletal and spinal COSs suggests a shared, cross-disciplinary recognition of their value. These domains not only reflect key patient priorities but also provide meaningful indicators of clinical progress and treatment efficacy. However, the review also identified significant gaps. Despite an increasing appreciation of spinal pain as a multifactorial and biopsychosocial condition, few COSs extended beyond traditional domains. Domains such as sleep disturbance, psychological wellbeing, social and environmental determinants, and physical, biological or physiological markers were infrequently represented. These omissions are important, as they suggest that many existing COSs may fall short of supporting the aims of precision rehabilitation or personalised medicine in spinal pain research and practice—a concern amplified by accumulating evidence that factors such as sleep disturbance, psychological wellbeing, and environmental influences significantly impact pain persistence, variability in treatment response, and long-term functional outcomes [9,64–67].

A growing body of evidence underscores the need to expand the scope of outcome measurement in spinal pain research to encompass currently underrepresented dimensions. Increasingly, studies have demonstrated that factors such as sleep quality, neuroimmune interactions, and psychological constructs—including resilience, catastrophising, and self-efficacy—influence the onset, progression, and treatment response of spinal pain [9,64–72]. Likewise, social and environmental determinants—such as levels of social support, histories of trauma, loneliness, access to healthcare services, and broader ecological exposures—are recognised as pivotal in shaping the pain experience and recovery trajectory, yet these remain largely absent from existing core outcome sets [6,12,20,73–76].

Beyond psychosocial and contextual domains, the integration of biological and physical metrics into COS development represents a sound direction for future development of holistic outcome sets. Biomarkers, neuroimaging findings, genetic and epigenetic indicators, and physical performance measures offer the potential to better characterise patient subgroups and tailor therapeutic approaches more precisely. Recent advances in omics technologies and wearable sensor data further enhance this potential by allowing real-time, individualised monitoring of health status and functional capacity [77–81]. Owing perhaps to many of the COSs in this review pre-dating the wide and growing accessibility of advanced performance and biological sensors, their absence appears to signal a need for planned regular updates to core sets. In this context, the results of our review can expedite such updates. It appears widely accepted that the four common domains reported herein should remain part of such COSs, enabling development groups to dedicate energies on appraising new technologies for their usefulness in updates sets.

To establish such comprehensive COSs, interdisciplinary collaboration is essential. Researchers from across clinical, social, psychological, and biological sciences must work collaboratively to ensure that COSs are sufficiently multidimensional to support the complexity of spinal pain assessment. Crucially, this includes not only healthcare professionals and academic researchers, but also patients and members of the public [82,83]. Patient and public involvement (PPI) is critical to ensure that the outcomes valued by those living with spinal pain are meaningfully represented [84]. Without this, outcome sets risk being overly researcher-centric, potentially overlooking aspects of health and well-being that patients consider important [83].

To enhance adoption and relevance across settings, COS implementation must be accompanied by a staged, evidence-informed process. First, robust psychometric appraisal of candidate instruments is needed to ensure reliability and validity across spinal populations. Second, feasibility testing in routine clinical environments (across primary, secondary, and tertiary care) is required to determine acceptability and integration potential. Finally, pilot implementation studies should be conducted to evaluate the COS's impact on clinical decision-making, patient experience, and research data quality. Without this staged approach, there is a risk that even well-constructed COSs may fail to achieve meaningful uptake or influence practice.

The findings of this review demonstrate that COSs developed for clinical practice were generally shorter and included fewer domains—primarily focusing on pain, function, and disability—to reduce burden during consultations. In contrast, COSs designed for research settings tended to incorporate a wider array of domains, including psychosocial, work-related, and quality-of-life measures, reflecting the need for more comprehensive data collection. This clear distinction in scope and content highlights the importance of differentiating COSs by intended use to ensure both feasibility in clinical settings and rigour in research contexts [85]. While overlap between the two is desirable, their purposes differ. Research COSs must enable standardisation and comparison across trials, support meta-analyses, and permit the detection of treatment effects. Clinical COSs, in contrast, must be feasible, efficient, and sensitive to the needs of individual patients. To date, few COSs have clearly articulated whether their intended use is in research, clinical care, or both [85]. This is an area requiring further clarification in future work.

## Strengths and limitations

A key strength of this systematic review lies in its methodological rigour and comprehensive scope, incorporating a wide range of core outcome sets (COSs) across multiple spinal regions and related pain presentations. The use of a clearly defined protocol, adherence to PRISMA guidelines, and robust data extraction and synthesis procedures contribute to the overall reliability and transparency of the review. However, an important limitation concerns the quality appraisal of the included studies. Many of the identified studies employed consensus-based methodologies, such as Delphi techniques, for which no universally accepted risk-of-bias tool currently exists. To address this challenge, we used the CREDES framework in combination with a modified version of the Buchbinder Appraisal Scale to evaluate methodological quality [40,43–45]. Nonetheless, a number of studies scored poorly on quality assessment, not necessarily due to poor conduct, but because they were undertaken prior to the development of formalised COS reporting and development standards, such as the core outcomes sets standards for design (COS-STAD) and reporting (COS-STAR) [41,42]. As a result, several studies lacked comprehensive reporting on key aspects of methodology, including the rationale for consensus methods, recruitment of expert panels, and stakeholder engagement. This limits the ability to fully appraise internal validity and may have introduced heterogeneity in study quality that could affect the interpretation of findings. Another methodological limitation of this review is the absence of a formal assessment of confidence in the findings. Frameworks such as GRADE and GRADE-CERQual, although robust in intervention and qualitative research settings respectively, were not applicable to the consensus-based studies included in this review [49,51]. This limits the ability to make strong recommendations about which domains or outcome measures should be universally adopted in future COS development and underscores the pressing need for a validated approach to assessing the certainty of evidence in COS development studies.

## Conclusion

This systematic review offers the first comprehensive synthesis of COSs currently used to assess spinal pain. It identifies consistent emphasis on core domains such as pain intensity, physical function, and participation in daily and work-related activities, reinforcing their relevance to both clinical practice and research. However, it also highlights fragmentation in outcome measurement and a lack of integration of broader biopsychosocial and biological domains. The existence of several COSs for a single body region (i.e., neck, low back) also poses challenges to the reason such sets exist, intended as a set

of core outcomes that are consistently collected across all relevant research studies. With no one COS for these regions being widely implemented, the intention of having a COS has yet to be realized in spinal pain. The common domains across sets may be a sound point of departure for the next iteration of spinal COSs towards widespread adoption.

The findings provide critical insights for researchers, policymakers, and healthcare leaders in selecting outcome measures that support rigorous, clinically safe, and economically sound research and evaluation. Importantly, this review underscores the need for a consensus-based, internationally recognised COS that reflects the multidimensional nature of spinal pain. Developing such a COS will improve study comparability, reduce research waste, enable robust meta-analyses, and guide personalised, high-quality care.

## Supporting information

**S1 File. Quality assessment.**
(DOCX)

**S2 File. PRISMA Checklist.**
(DOCX)

## Author contributions

**Conceptualization:** Tim Noblet, David Walton, Alison Rushton.

**Data curation:** Tim Noblet, Carol Li, Richard Newsham-West, David Walton, Alison Rushton.

**Formal analysis:** Tim Noblet, Alison Rushton.

**Investigation:** Tim Noblet.

**Methodology:** Tim Noblet, Carol Li, David Walton, Alison Rushton.

**Project administration:** Tim Noblet.

**Writing – original draft:** Tim Noblet.

**Writing – review & editing:** Tim Noblet, Carol Li, David Walton, Alison Rushton.

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
