## [Decision Letter · Decision Letter 0]

13 Oct 2025

Dear Dr. Noblet,

Thank you for submitting your manuscript to PLOS ONE. After careful consideration, we feel that it has merit but does not fully meet PLOS ONE’s publication criteria as it currently stands. Therefore, we invite you to submit a revised version of the manuscript that addresses the points raised during the review process.

**Authors are required to reply all the queries, raised by the reviewers.**

We look forward to receiving your revised manuscript.

Kind regards,

Priti Chaudhary, M.S.

Academic Editor

PLOS ONE

**Journal Requirements:**

1. When submitting your revision, we need you to address these additional requirements. Please ensure that your manuscript meets PLOS ONE's style requirements, including those for file naming. The PLOS ONE style templates can be found at https://journals.plos.org/plosone/s/file?id=wjVg/PLOSOne_formatting_sample_main_body.pdf and https://journals.plos.org/plosone/s/file?id=ba62/PLOSOne_formatting_sample_title_authors_affiliations.pdf 2. We note that your Data Availability Statement is currently as follows: All relevant data are within the manuscript and its Supporting Information files. Please confirm at this time whether or not your submission contains all raw data required to replicate the results of your study. Authors must share the “minimal data set” for their submission. PLOS defines the minimal data set to consist of the data required to replicate all study findings reported in the article, as well as related metadata and methods (https://journals.plos.org/plosone/s/data-availability#loc-minimal-data-set-definition). For example, authors should submit the following data: - The values behind the means, standard deviations and other measures reported;- The values used to build graphs;- The points extracted from images for analysis. Authors do not need to submit their entire data set if only a portion of the data was used in the reported study. If your submission does not contain these data, please either upload them as Supporting Information files or deposit them to a stable, public repository and provide us with the relevant URLs, DOIs, or accession numbers. For a list of recommended repositories, please see https://journals.plos.org/plosone/s/recommended-repositories. If there are ethical or legal restrictions on sharing a de-identified data set, please explain them in detail (e.g., data contain potentially sensitive information, data are owned by a third-party organization, etc.) and who has imposed them (e.g., an ethics committee). Please also provide contact information for a data access committee, ethics committee, or other institutional body to which data requests may be sent. If data are owned by a third party, please indicate how others may request data access. 3. We notice that your supplementary files are included in the manuscript file. Please remove them and upload them with the file type 'Supporting Information'. Please ensure that each Supporting Information file has a legend listed in the manuscript after the references list. 4. If the reviewer comments include a recommendation to cite specific previously published works, please review and evaluate these publications to determine whether they are relevant and should be cited. There is no requirement to cite these works unless the editor has indicated otherwise. 

Reviewers' comments:

**Comments to the Author**

1. Is the manuscript technically sound, and do the data support the conclusions?

Reviewer #1: Yes

Reviewer #2: Yes

2. Has the statistical analysis been performed appropriately and rigorously?

Reviewer #1: I Don't Know

Reviewer #2: N/A

3. Have the authors made all data underlying the findings in their manuscript fully available?

Reviewer #1: Yes

Reviewer #2: Yes

4. Is the manuscript presented in an intelligible fashion and written in standard English?

Reviewer #1: Yes

Reviewer #2: Yes

**Reviewer #1: ** Thanks for giving the opportunity to review this valuable work, I read the full manuscript. Overall, the study is timely, and the methods are sensible, but the manuscript needs specific edits for clarity.

Abstract

Percentages are reported but the denominator is not shown. Always report n (%) so readers can interpret percentages correctly.

While recommending a Delphi study is sensible, the abstract should present this as a logical next step rather than a definitive solution. The authors should also acknowledge potential challenges to such a process.

Introduction

The first two paragraphs (lines ~66–89) both describe global burden, heterogeneity, and costs. These could be combined and streamlined for concision.

The figure “790 million people” should be tied to specific Global Burden of Disease years/metrics. Readers expect exact sources and dates.

The Introduction notes lack of consensus on a single COS but does not sufficiently review prior COS efforts. A brief overview of past efforts would help clarify novelty.

Some method-like statements in the Introduction such as breadth of research approaches and study types could be trimmed or relocated to Methods. The Introduction should stay focused on background, gap, and aim.

Methods

The Methods do not state whether non-English records were eligible.

Discussion

Much of the Discussion re-iterates results rather than synthesizing them into deeper interpretation. The authors should move beyond “what” was found to explain why patterns exist and what that means for implementation.

There’s a missed chance to present concrete, short-term recommendations for researchers.

Beyond identifying domains, a staged process is needed: review psychometric evidence for candidate instruments, test feasibility across settings, and pilot implementation before finalizing a minimal core set.

**Reviewer #2:**  The manuscript was interesting and good written and discussed.

Introduction was good written.

Materials and methods were good designed.

Results were good described.

Discussion was good written and include the conclusion of the study. .

**Do you want your identity to be public for this peer review?** For information about this choice, including consent withdrawal, please see our Privacy Policy

Reviewer #1: No

Reviewer #2: No

---

## [Author Response · Author response to Decision Letter 1]

17 Nov 2025

Please see attached response to reviewer doc

---

## [Editor Report · Decision Letter 1]

19 Nov 2025

Core Outcome Sets for Spinal and Associated Limb, Trunk, Abdomen or Pelvic Pain: a systematic review

PONE-D-25-44505R1

Dear Dr. Tim Noblet,

We’re pleased to inform you that your manuscript has been judged scientifically suitable for publication and will be formally accepted for publication once it meets all outstanding technical requirements.

Kind regards,

Priti Chaudhary, M.S.

Academic Editor

PLOS ONE
---

## [Editor Report · Acceptance letter]

PONE-D-25-44505R1

PLOS ONE

Dear Dr. Noblet,

I'm pleased to inform you that your manuscript has been deemed suitable for publication in PLOS ONE. Congratulations! Your manuscript is now being handed over to our production team.

Kind regards,

on behalf of

Dr. Priti Chaudhary

Academic Editor

PLOS ONE